# Comparative Study of Silk Fibroin-Based Hydrogels and Their Potential as Material for 3-Dimensional (3D) Printing

**DOI:** 10.3390/molecules26133887

**Published:** 2021-06-25

**Authors:** Watcharapong Pudkon, Chavee Laomeephol, Siriporn Damrongsakkul, Sorada Kanokpanont, Juthamas Ratanavaraporn

**Affiliations:** 1Biomedical Engineering Program, Faculty of Engineering, Chulalongkorn University, Bangkok 10330, Thailand; w.pudkon@gmail.com; 2Biomedical Engineering for Medical and Health Research Unit, Faculty of Engineering, Chulalongkorn University, Bangkok 10330, Thailand; papomchavee@gmail.com (C.L.); Siriporn.D@chula.ac.th (S.D.); Sorada.K@chula.ac.th (S.K.); 3Department of Chemical Engineering, Faculty of Engineering, Chulalongkorn University, Bangkok 10330, Thailand; 4Biomedical Engineering Research Center, Faculty of Engineering, Chulalongkorn University, Bangkok 10330, Thailand

**Keywords:** silk fibroin, hydrogel, material for 3D-printing, 3D-printing

## Abstract

Three-dimensional (3D) printing is regarded as a critical technology in material engineering for biomedical applications. From a previous report, silk fibroin (SF) has been used as a biomaterial for tissue engineering due to its biocompatibility, biodegradability, non-toxicity and robust mechanical properties which provide a potential as material for 3D-printing. In this study, SF-based hydrogels with different formulations and SF concentrations (1–3%wt) were prepared by natural gelation (SF/self-gelled), sodium tetradecyl sulfate-induced (SF/STS) and dimyristoyl glycerophosphorylglycerol-induced (SF/DMPG). From the results, 2%wt SF-based (2SF) hydrogels showed suitable properties for extrusion, such as storage modulus, shear-thinning behavior and degree of structure recovery. The 4-layer box structure of all 2SF-based hydrogel formulations could be printed without structural collapse. In addition, the mechanical stability of printed structures after three-step post-treatment was investigated. The printed structure of 2SF/STS and 2SF/DMPG hydrogels exhibited high stability with high degree of structure recovery as 70.4% and 53.7%, respectively, compared to 2SF/self-gelled construct as 38.9%. The 2SF/STS and 2SF/DMPG hydrogels showed a great potential to use as material for 3D-printing due to its rheological properties, printability and structure stability.

## 1. Introduction

Three-dimensional-printing technology has attracted considerable attention as a promising tool for tissue engineering and regenerative medicine [1]. Structural-complex scaffolds can be precisely designed using software and fabricated in a high resolution using a layer manufacture. The development of printable material is one of the critical parts in 3D-printing researches [2]. The critical features required for printable materials are biocompatibility, bioactivity, biodegradation and mechanical stability [3,4]. In addition, the material should be able to encapsulate cells and maintain cell viability for long-term tissue culture. Natural-derived polymers are attractive candidates to be applied as bioink.

Silk fibroin (SF) is a naturally derived fibrous protein produced by domesticated *Bombyx mori* silkworms. Due to its characteristics, including biocompatibility, robust mechanical properties, biodegradability, sterilizability, high thermal stability and microbial resistance, SF has been widely studied in biomedical fields [5,6]. The primary structure of SF contains a large number of glycine–alanine repetitive sequences which are accounted for the formation of thermodynamically stable β-sheet structure. Therefore, comparing to other natural polymers, SF possesses a slower degradation rate as well as higher mechanical robustness [7]. From our previous studies, SF-based materials were developed and applied for various applications such as bone tissue regeneration, wound healing and drug-controlled release systems [8,9,10]. Moreover, the cytotoxicity of SF-based bone scaffold has been proved according to ISO 10993 [11]. Different techniques have been reported for fabrication of various Thai SF formats such as film [12], scaffold [13], tube [14], sponge [15], microsphere [16], fiber [17] and hydrogel [18,19,20]. For 3D-printing, SF-based hydrogels showed an interesting feature which could be appropriate to serve as the material for 3D-printing, e.g., high water content, shape plasticity, suitable rheological properties, low surface tension and the ability of crosslinking [21].

Generally, SF hydrogels can be spontaneously formed from the regenerated SF solution within a couple of weeks to months by self-assembly processes [22]. The long gelation time of self-gelled SF may limit its application. Different strategies were introduced to induce the rapid gelation of SF. The gelation of SF through enzymatic crosslinking can be induced using enzymes, such as horseradish peroxidase or transglutaminase, in which the gelation can occur under physiological conditions, allowing the cell encapsulation during the gelation [23]. Moreover, photopolymerization, and the chemical crosslinkers, such as carbodiimides, glutaraldehyde, or genipin were used. Our previous studies demonstrated various formulations of SF-based hydrogels that can be induced by different agents, such as alcohols [24], an anionic surfactant, sodium tetradecyl sulfate (STS) [25], and a phospholipid 1,2-dimyristoyl-*sn*-glycerol-3-phospho-(1′-*rac*-glycerol) (DMPG) [26]. STS is as anionic surfactant that have been shown to accelerate the gelation of SF rather than the cationic and nonionic surfactant. Moreover, STS has been used in medical products approved by the U.S. Food and Drug Administration (US-FDA). DMPG is a phosphorylglycerol with negative charged amphipathic lipids. It has the ability to induce conformational changes in several water-soluble proteins due to electrostatic and hydrophobic interactions. For STS and DMPG, the gel formation time ranged from 20 min to less than an hour depending on the amount of the additives and the gelation mechanisms. The electrostatic and hydrophobic interactions between SF and either STS or DMPG induced the structural transition from the amorphous random coil to the stable β-sheet structure, leading to the gel formation. In addition, the cytocompatibility and in vitro degradation of SF/STS and SF/DMPG hydrogels were also demonstrated. Our preliminary study on the application of these SF-based hydrogels in 3D-printing have been performed. Interestingly, we found that SF-based hydrogels induced by STS and DMPG tended to have printability potential. Therefore, it is worth exploring the properties and printability of these SF-based hydrogels and introducing them as material for 3D-printing applications. 

In this study, the gelation time, rheological properties, printability and post-treatment processes, and chemical structure of SF-based hydrogels induced by STS and DMPG were systematically investigated in comparison to those of the self-gelled SF hydrogel. The information from this study would be useful for the selection of suitable SF-based hydrogels as material for 3D-printing application. Moreover, this study is the first report applying the STS and DMPG-induced SF hydrogels in 3D-printing technology.

## 2. Materials and Methods

### 2.1. Preparation of Regenerated Silk Fibroin Solution

Thai *Bombyx mori* silk cocoons (Nangnoi Srisaket 1) were kindly supplied from Queen Sirikit Sericulture Center, Nakhon Ratchasima, Thailand. The regenerated aqueous SF solution was prepared following an established protocol [27]. Firstly, silk cocoons were boiled in 0.02 M sodium carbonate (Na_2_CO_3_) for 20 min to remove sericin and then washed with deionized water before leaving to dry. Subsequently, the degummed silk fibers were dissolved in 9.3 M lithium bromide (LiBr) at 60 °C for 4 h. The SF solution was then dialyzed against deionized water for 3 days using dialysis membrane (Molecular weight cut-off = 12,000–16,000 Da, Vikase Company Inc., Osaka, Japan) to remove salt ions [28]. Then, the dialyzed SF solution was centrifuged to remove impurities. The final concentration of the obtained aqueous SF solution was approximately 6–7%wt. The regenerated SF solution was stored at 4 °C until used.

### 2.2. Preparation of SF-Based Hydrogels

Three formulations of SF hydrogels including natural gelation (SF/self-gelled), STS-induced (SF/STS) and DMPG-induced (SF/DMPG) were prepared according to the methods reported in our previous researches with slight modifications (Table 1) [25,26]. The SF solution was loaded in the syringe and the hydrogel was formed within the syringe used for 3D-printing. The final concentrations of SF hydrogels were fixed at 1%, 2% and 3%wt while the optimal concentrations STS and DMPG were selected from previous studies. For SF/STS hydrogel, the final concentration of STS was fixed at 0.09% *w*/*v* with a ratio of SF to glycerol at 3:1 (*w*/*w*). For SF/DMPG hydrogel, the final concentration of DMPG was fixed at 0.35% *w*/*v*. As a control, the SF/self-gelled hydrogel was prepared by incubation of SF solution at 60 °C until complete gelation was occurred. After the complete gelation, all hydrogels were further incubated at 60 °C for 24 h before 3D-printing process.

### 2.3. Gelation Time Determination

The mixed solutions of SF and STS or DMPG were prepared as previously described at different concentrations of SF (1%, 2% and 3%). The gelation time was determined from the turbidity change by a measure of optical density [28]. The measurement was monitored at 550 nm using a Microplate Reader (FLUOstar Omega, Thermo Fisher Scientific, MA, USA) at 37 °C. Then, the gelation time was defined as the point where the average optical density reached as half-maximum value.

### 2.4. Measurement of Rheological Properties 

The Haake Mars Rheometer (Thermo Fisher Scientific, Massachusetts, USA) with temperature maintained at 37 °C was used for all experiments. A 35 mm parallel plate geometry was used with a gap value of 1 mm. At the first step, a time-sweep experiment was conducted by applying 0.5% strain and a frequency of 1 Hz. This test measures complex modulus change after SF solution has been loaded as time advances. Secondly, a frequency-sweep experiment was carried out with an applied strain of 0.5% and the frequency ranged from 0.5 to 100 Hz to determine the mechanical stability of the hydrogels. For the third step, the shear-thinning behavior and structure recovery of SF hydrogel were measured using a thixotropic analysis. These properties were analyzed by obtaining cycle through a 2-step experiment including up curve by increasing shear rate as 100 s^−1^ for 2 min and down the curve by decreasing shear rate to 0.1 s^−1^ for 1 min and repeated for more 2 cycles. From the experiment data, the degree of structure recovery (% recovery) was calculated according to the following Equation (1)
(1)$$\%recovery=\frac{\eta_{2}}{\eta_{1}}\times 100$$
where ƞ_1_ represents the average viscosity of hydrogel at the first cycle before applied shear rate, and ƞ_2_ represents the average viscosity of hydrogel at the second cycle after applied shear rate as 100 s^−1^.

### 2.5. 3D-printing and Post-treatment Processes

Before 3D-printing, all SF-based hydrogel formulations were prepared and gelled in a syringe and then connected to 3D Bioplotter (3D-Bioplotter^®^ Manufacturer Series, EnvisionTEC, Gladbeck, Germany). To fabricate the printed construct, the printing parameters as extruded pressure and movement speed were optimized for each SF-based hydrogel formulations (Table 2). The SF hydrogels were extruded through the nozzle with diameter as 0.41 mm. The construct design was made using the Perfactory RP software (EnvisionTEC) by converting designed part file (STL) file to printing code (G-code). The 3D-structure was printed as a 30 × 30 × 5 mm^3^ box with a 4-layer thickness. The inner pattern was printed in an alternating pattern, in which each one was aligned 90° from the layer below it. The printed hydrogels obtained were placed on the Petri dish for further post-treatment processes.

For post-treatment processes, the printed hydrogels were crosslinked by UV irradiation for 20 min at room temperature, then dried at room temperature for 4 h and followed by 70%wt ethanol immersion for 2 h. After that, the printed hydrogels were washed and stored in DI water until further tests.

### 2.6. Secondary Structure Analysis

Secondary structures of SF-based hydrogels were determined through Fourier Transform Infrared Spectroscopy (FTIR, using IRPrestige 21, Shimadzu, Kyoto, Japan) in an attenuated total reflection (ATR) mode. The lyophilized samples were finely ground and casted onto the ZnSe cell, before collecting the spectrum from 4000 to 800 cm^−1^ with 2.0 cm^−1^ resolution and 1 cm^−1^ interval. The chemical structure of the SF solution, SF/STS and SF/DMPG mixtures, the printed hydrogel before and after post-treatment were evaluated. 

To quantify the secondary structure, Fourier self-deconvolution and curve fitting of the infrared spectra were performed. The deconvolution of the amide I region (1575–1725 cm^−1^) were conducted using Omnic 8.0 software. The Voigt line shape with a half bandwidth of 10 cm^−1^ and an enhancement factor of 3.0 were applied. The curve fitting of the deconvoluted spectrum was performed using Origin Pro 9.0 software. The amount of β-sheet conformation was calculated from the sum of the percentage of area under the peaks in the 1616 to 1637 cm^−1^ and 1696 to 1703 cm^−1^ regions. The amount of random coil (1638–1655 cm^−1^), alpha helix (1656–1662 cm^−1^), and β-turn (1663–1696 cm^−1^) structures were also calculated from the area under the defined peaks.

## 3. Results 

### 3.1. Gelation Time of SF-Based Hydrogels

Gelation times of SF-based hydrogels were shown in Table 1. The addition of STS and DMPG in the SF solution significantly accelerated the gelation. It can be observed that the gelation time of both SF/STS and SF/DMPG hydrogels decreased with an increasing SF concentration. For SF/STS formulation, the 3SF/STS and 2SF/STS hydrogels were formed within 19 and 36 min, respectively, while the gelation of 1SF/STS was not completed within 2 h. For SF/DMPG formulation, the gelation time of 1SF/DMPG, 2SF/DMPG and 3SF/DMPG were 96 min, 13 min and 8 min, respectively. The gelation time of SF-based hydrogel induced with DMPG was faster than inducing with STS comparing the similar SF concentration. On the other hand, the spontaneous gelation of regenerated SF solution took over two weeks; therefore, the gelation time of SF/self-gelled cannot be measured.

### 3.2. Rheological Properties of SF-Based Sol-Gel

The time-sweep measurement of SF-based hydrogels (Figure 1a–c) reported the storage modulus (G’) and loss modulus (G”) over time. For SF/STS and SF/DMPG, the solution possessed a fluid-like behavior in the beginning as G’ and G” were not clearly distinguishable. Then, both G’ and G” of the hydrogels increased before maintain a plateau at their respective gelation time. The G’ was higher than the G” for all formulations that was the characteristic of formation from sol-state to gel-state. However, in case of SF/self-gelled, the completed hydrogel was used to measure due to its long gelation time, therefore, the G’ was higher than G” with constant value from an initial time. 

The frequency-dependent behavior of SF-based hydrogels was assessed using oscillatory frequency sweeps as shown in Figure 1d–f. For all SF-based hydrogels, the storage modulus was constant within the frequency range of 0.1–10 Hz, before becoming fluctuated when applied the frequency was over 10 Hz. The storage modulus of 1SF/self-gelled, 2SF/self-gelled and 3SF/self-gelled were 332.5, 2739.9 and 10,614.2 Pa, respectively. For SF/STS formulation, the storage modulus showed 52.2, 790.7 and 2294.4 Pa for 1SF/STS, 2SF/STS and 3SF/STS, respectively. For SF/DMPG formulation, the storage modulus of 1SF/DMPG, 2SF/DMPG and 3SF/DMPG were 9.7, 510.2 and 1471.1 Pa, respectively. The storage modulus increased with increasing SF concentration for all SF-based hydrogel formulations. At the same SF concentration, the storage modulus of SF/self-gelled hydrogel was highest while the lowest storage modulus exhibited in SF/DMPG hydrogels.

The shear-thinning behavior and structure recovery of SF-based hydrogels measured by thixotropic analysis were shown in Figure 2. All SF-based hydrogels showed shear thinning behavior with different degrees of structure recovery. An initial viscosity of SF-based hydrogel increased with increasing SF concentration for all formulations. When a shear rate of 100 s^−1^ was applied, the viscosity of all SF-based hydrogels decreased rapidly. After the shear rate was decreased to 0.1 s^−1^, the viscosity of SF-based hydrogels was subsequently recovered closely to their initial value. The structural recovery could be observed at every cycle of shear rate changed. Table 2 demonstrated the calculated recovery percentage of SF-based hydrogels in the resting stage after the high shear rate was applied. The degree of structure recovery of SF/self-gelled increased with an increasing of SF concentrations. For SF/STS and SF/DMPG formulations, the 2SF-based hydrogels showed the highest degree of structure recovery. The highest percentage of structural recovery of each formulation can be observed as 44.2%, 70.4% and 53.7% for 3SF/self-gelled, 2SF/STS and 2SF/DMPG, respectively. As the result, the SF-based hydrogels induced with STS and DMPG exhibited higher degree of structure recovery than SF/self-gelled formulation.

### 3.3. Printability 

Printability of SF-based hydrogels was assessed by printing a 4-layer construct of box model. From the preliminary study, a clogging of hydrogel at the nozzle was observed when 3SF-based hydrogels were printed. Consequently, the 2SF-based hydrogels were chosen for extrusion from 3D-bioprinter. The 2SF-based hydrogels were printed and crosslinked to stabilize the printed constructs as shown in Figure 3. During printing, the temperature of the syringe was maintained at 37 °C. The SF-based hydrogels were extruded on the Petri dish at the optimized printing conditions. The deposition temperature was set at 25 °C. Low deposition temperature helped to stabilize the structure of printed constructs by preventing the spread or deformation of printed hydrogel. All 2SF-based hydrogels could be printed into a 4-layer box model without collapse. Subsequently, the printed constructs were crosslinked under UV-irradiation for 20 min, air dried at room temperature for 4 h, and immersion in 70%wt ethanol for 2 h. After post-treatment processes, it can be observed that the structures of printed constructs from 2SF/STS and 2SF/DMPG hydrogels were maintained, while those of 2SF/self-gelled hydrogel were partially deformed. 

### 3.4. Chemical Structure and Secondary Conformation of SF-Based Hydrogels

The chemical structures of the 2SF/self-gelled, 2SF/STS and 2SF/DMPG constructs after printed and post-treatment were characterized through ATR-FTIR technique using 2SF solution as a control (Figure 4). The spectra of 2SF-based hydrogels showed the characteristic peaks of protein including amide I, II and III at 1650, 1550 and 1300 cm^−1^, respectively. The characteristic peaks of CH_3_ and CH_2_ stretching at 2955, 2873 and 2849 cm^−1^ were clearly observed on the spectra of 2SF/STS and 2SF/DMPG formulations due to the structure of inducing agents. The FTIR spectra were deconvoluted of each secondary conformation of the hydrogels after printed and after post-treatment were shown in Figure 4d. All SF-based hydrogels exhibited higher content of β-sheet compared to SF solutions. To compare the β-sheet content between after printed and after post-treatment, the β-sheet content increased while the random coil content seemed to decrease. After post-treatment, the 2SF/DMPG constructs showed higher β-sheet content, as 77.2 %, than 2SF/STS and 2SF/self-gelled, as 68.7 and 72.1%, respectively.

## 4. Discussion

In general, the regenerated SF solution can turn gel through a self-assembly process by a chain rearrangement and hydrogen bonding that produce the transformation of random coil to a stable β-sheet form [29]. This gelation processes takes over two weeks, which is not practicable for the applications as cell-encapsulated bioink or injectable hydrogels. To reduce the gelation time, some amphipathic chemicals, such as surfactants and phospholipids were added into SF solution [25,26]. It can enhance the formation of β-sheet structure due to electrostatic and hydrophobic interactions. In this study, the mixtures of SF with STS or DMPG showed the complete gelation within about 8-96 min while the SF/self-gelled formulations took approximately more than 2 weeks. Moreover, the gelation time of SF solution depends on SF concentration that a higher SF concentration resulted in a shorter gelation time. In addition, the mixture of glycerol also reduced the gelation time of SF/STS hydrogels due to the polarity and hydrophobicity [24]. Moreover, glycerol can act as a physically stabilized agent that beneficial for the extrusion of SF/STS hydrogels [30]. When SF mixed with glycerol, the homogeneity of SF/STS hydrogel was enhanced, and the printable hydrogel was obtained. 

The shear-thinning properties as well as the structure recovery of the printing materials are required to confirm the applicability for material which the SF-based hydrogels should be extruded from the nozzle, encounter the high shear stress, and able to recover the initial structure immediately after finishing the printing process [31]. From Figure 1, the storage modulus of the hydrogels increased with an increasing of the SF concentration. These could imply that SF hydrogels with lower SF concentration required a lower pressure for extrusion and would be easier to be printed comparing to the stiffer hydrogels with a higher SF concentration. It is noted that the storage modulus of SF/self-gelled hydrogel was higher than SF/STS and SF/DMPG hydrogels. In general, the regenerated SF solution can turn gel through a self-assembly process by a chain rearrangement and hydrogen bonding to transform random coil to a stable beta sheet structure. With the long incubation time, the SF structure can crystallize and form more stable structure, resulting in improved strength of SF hydrogel. When the additives such as STS and DMPG are mixed with the SF solution, it can enhance the formation of beta sheet structure due to the electrostatic and hydrophobic interactions. Adding gel enhancing chemicals also interferes with the hydrogen bond between the protein and water due to the formation of hydrogen bond in their structures and the steric hindrance effect [22]. Thus, mixture of STS and DMPG might lead to a decrease in the hydrogen bonds, resulting in a lower storage modulus of SF/STS and SF/DMPG compared to SF/self-gelled hydrogel. Furthermore, the elastic behavior was predominated over the viscous nature in the frequency range of 0.1-10 Hz. It could be explained that all SF-based hydrogel formulations had the gel-state in this frequency range. For this study, it found that 1SF-based hydrogels would not be mechanically sufficient to maintain their architectural structure upon extrusion due to the lowest storage modulus and the lowest degree of structure recovery. On the other hand, there was a clogging of hydrogel at the nozzle when using 3SF-based hydrogels even printed at high pressure. In the case of 2SF-based hydrogel, it showed suitable storage modulus for extrusion without clogging behavior. Consequently, 2SF-based hydrogel was chosen for a single-syringe injectable material for 3D-printing. From the previous report, the highest cell viability upon printing is observed when a low polymer content hydrogel is used in combination with low pressure [32]. Therefore, the SF concentration and extruded pressure for 2SF-based hydrogels could be beneficial for cell-laden and cell viability during printing process. Moreover, these hydrogels exhibited high degree of structure recovery and shear-thinning behavior which can allow ready flow of the fluid through confined nozzle diameters of the bioprinter. At the same time, immediately after extrusion, the fluid can demonstrate instant shape stability to assemble a 3D-structural assembly without collapse. The printing parameters were optimized for each 2SF-based hydrogel formulations. The 2SF/self-gelled formulation required higher extruded pressure than 2SF/STS and 2SF/DMPG due to the higher storage modulus. During the printing process, 2SF-based hydrogel would be attained because of a high maintaining the shape of the as-printed and a less pressure required. After printed, there was no evidence of structural collapse of the 4-layer box structure. This supports flow behavior of 2SF-based hydrogel that can extrude in a filament rather than a droplet during 3D-printing. Then, the printed structure was treated by three-step post-treatment. Initially, the printed hydrogels were irradiated by UV light which was proposed to enhance the crosslinking of SF molecules. Generally, the water molecules presented in the SF hydrogel could be intrigued by UV and could produce a large number of free radicals. The aromatic side chains of tyrosine and phenylalanine in the SF would be attacked by the free radicals, leading to the bond formation [33]. However, only the UV treatment was not enough to crosslink and stabilize the printed structures, then another treatment step was required. Therefore, the post-treatment with ethanol immersion was conducted. The ethanol immersion resulted in an immediate effect on the secondary structures by diffusion into the printed hydrogels and induced β-sheet formation [34,35]. However, the deformation of the printed 2SF/self-gelled constructs was observed after ethanol immersion for 2 h. It can be explained that the degree of structure recovery for 2SF/self-gelled was lower than 2SF/STS and 2SF/DMPG. Consequently, the printed structure was easily broken from diffusion of ethanol into the printed structure. Additionally, the induced β-sheet content after three-step post-treatment was investigated by ATR-FTIR technique by compared between after printed and after post-treatment processes using 2SF solution as control. There are three structures in which silk fibroin can exist: silk I, II and III, where silk I is a structural form consisting of more water and less β-sheet content, silk II is a structural form consisting of β-sheet and silk III is a three-fold helical structure that is observed at the air-water interface [36]. Furthermore, the β-sheet content of 2F-based hydrogel increased after post-treatment processes compared to after printed. It can be seen that the structure transformation to stable form with enhanced structure stability was occurred through the three-step post-treatment. From our results, the 2SF/STS and 2SF/DMPG hydrogels showed better flow behavior, higher structure recovery and higher structure stability than 2SF/self-gelled hydrogel during printing and after post-treatment processes. In addition, the non-cytotoxicity of SF/STS and SF/DMPG hydrogel were shown in previous studies [26,37]. Therefore, these SF-based hydrogel formulations demonstrated the potential as material for 3D-printing and applied for several biomedical applications, such as tissue engineering.

## 5. Conclusions

In summary, SF-based hydrogel with different formulations were studied their properties to use as material for 3D-printing. The SF hydrogel with inducing agents as STS and DMPG can reduce the gelation time due to the electrostatic and hydrophobic interactions. Each SF-based hydrogel formulations were optimized by SF concentration to generate materials optimal under printing conditions. We found that the hydrogel with 2%wt SF was optimal for extrusion into the 4-layer box construct in all formulations without structural collapse. The three-step post-treatment can enhance the structure stability via induced β-sheet content of printed structure. The 2SF/STS and 2SF/DMPG hydrogels exhibited a good flow behavior without the nozzle clogging during the printing process. In addition, these hydrogels showed a high degree of structure recovery after printing and a high structure stability after post-treatment processes. Therefore, our results showed that SF-based hydrogels have the potential properties to be used as material for 3D-printing.

## Figures and Tables

**Figure 1 molecules-26-03887-f001:**
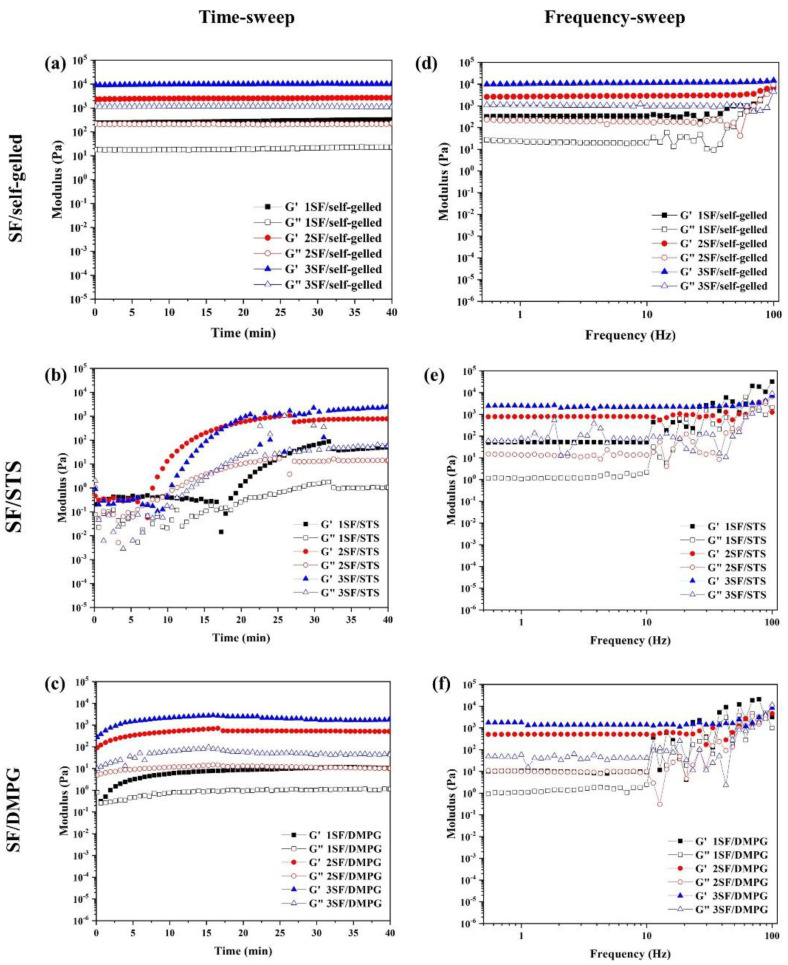
Time-sweep (**a**–**c**) and frequency-sweep (**d**–**f**) experiments showing storage (G’) and loss (G”) modulus of SF-based hydrogels with different SF concentrations.

**Figure 2 molecules-26-03887-f002:**
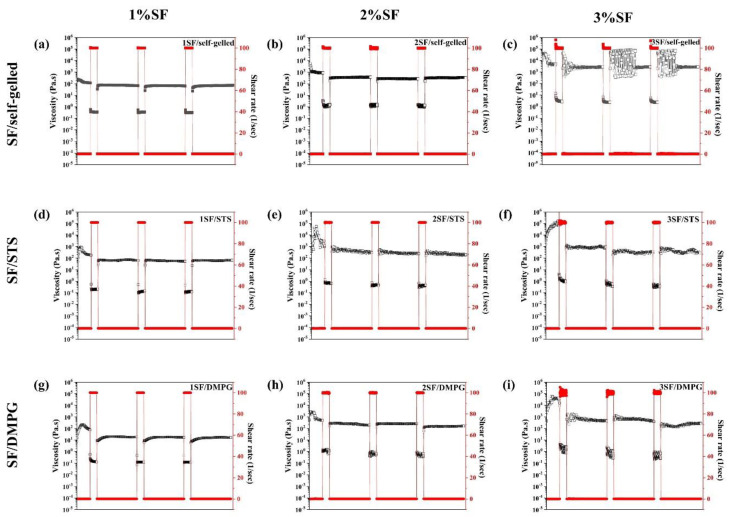
Viscosity of (**a**–**c**) SF/self-gelled, (**d**–**f**) SF/STS and (**g**–**i**) SF/DMPG hydrogels when applying different shear rate, indicating shear thinning behavior and structural recovery of the hydrogels.

**Figure 3 molecules-26-03887-f003:**
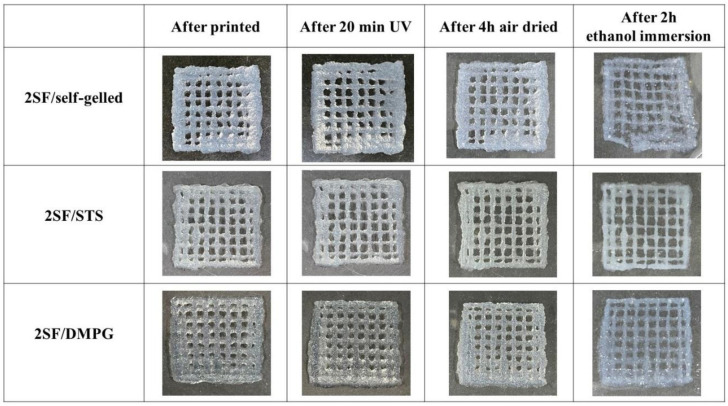
Structure of 4-layer box model constructs printed from 2SF-based hydrogels before and after post-treatment at each condition.

**Figure 4 molecules-26-03887-f004:**
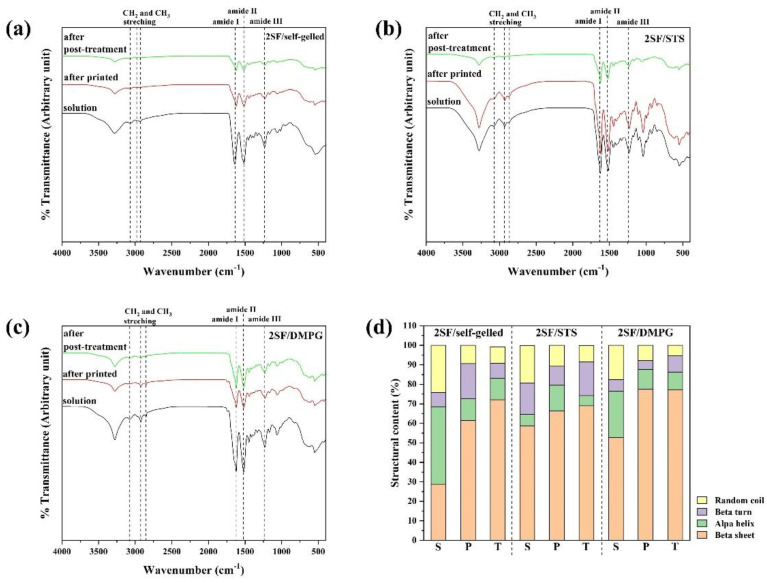
ATR-FTIR spectra of SF-based hydrogels (**a**) 2SF/self-gelled, (**b**) 2SF/STS, (**c**) 2SF/DMPG and (**d**) Percentage of secondary conformation of the constructs after printed (P) and post-treatment (T), deconvoluted from the amide I region of spectra using SF solution (S) as a control.

**Table 1 molecules-26-03887-t001:** Formulation, Nomenclature and Gelation Time of SF-based Hydrogels.

Name	%SF	Conc. of STS or DMPG	Ratio of SF/Glycerol (*w*/*w*)	Gelation Time
1SF/self-gelled	1%	N/A	N/A	>2 weeks
2SF/self-gelled	2%	N/A	N/A	>2 weeks
3SF/self-gelled	3%	N/A	N/A	>2 weeks
1SF/STS	1%	0.09%*w*/*v* STS	3:1	>120 min
2SF/STS	2%	0.09%*w*/*v* STS	3:1	36 min
3SF/STS	3%	0.09%*w*/*v* STS	3:1	19 min
1SF/DMPG	1%	0.35%*w*/*v* DMPG	N/A	96 min
2SF/DMPG	2%	0.35%*w*/*v* DMPG	N/A	13 min
3SF/DMPG	3%	0.35%*w*/*v* DMPG	N/A	8 min

N/A is not available.

**Table 2 molecules-26-03887-t002:** Printing parameters and degree of structure recovery of SF-based hydrogels.

Formulations	Speed (mm/s)	Pressure (bars)	Structure Recovery (%)		
			1%	2%	3%
SF/self-gelled	25	0.5	28.9%	38.9%	44.2%
SF/STS	15	0.3	33.2%	70.4%	1.4%
SF/DMPG	15	0.3	8.8%	53.7%	17.6%

## Data Availability

Data sharing is not applicable to this article.

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
