# Peer review of "Comparative Study of Silk Fibroin-Based Hydrogels and Their Potential as Material for 3-Dimensional (3D) Printing"

_molecules, 2021, doi:10.3390/molecules26133887_

Round 1
Reviewer 1 Report
Please see the attachment.

Reviewer 2 Report
Page 2, line 56 – 71: What about those studies using enzymes like Horseradish peroxidase or transglutaminase, photopolymerization or chemicals like Glutaraldehyde, Genipin or Carbodiimide reaction to induce silk fibroin gelation (as reviewed in Agostinacchio et al 2020 in Trends in Biotechnology, https://doi.org/10.1016/j.tibtech.2020.11.003). For a proper introduction of the scientific problem to all readers, those studies need to be stated. It is also important that the authors explain why they only focus on STS and DMPG. Is there any scientific rationale for those two components?
Page4, line 128: Chapter 2.5 is termed “D bioprinting…..”. Probably there is a “3” missing. However, in this context the authors should reconsider if the process is 3D bioprinting or rather 3D printing because they do not use a bioink. They only print a biomaterial. Bioprinting means printing of bioinks and bioinks need cells inside (see Groll et al. 2019 in Biofabrication, https://doi.org/10.1088/1758-5090/aaec52).
In the introduction (page 2, line 70/71) the authors claim Thai SF-based hydrogels are suitable for the use as bioinks. Furthermore, in the conclusion (page 11, line 328/329) the authors summarize that Thai SF-based hydrogels were studied to us as a bioink. In addition, their last conclusion (page 11, line 339) is that Thai SF-based hydrogels have potential to use as bioink. Unfortunately, they do not show that their SF/STS or SF/DMPG hydrogels are biocompatible and thus those hydrogels are suitable to carry living cells, as it is mandatory for a bioink. If the authors do not present cytocompatibility data of their “bioinks” then they have to change the conclusion. They only present data, which shows that their modified Thai SF-based hydrogels are suitable for use in 3D printing.
Page 10, line 324-326: The authors stated that Thai SF-based hydrogel formation could be used for biomedical applications, such as tissue engineering. During their studies, the authors did not show any data if their printed 3D structures are stable in water or cell culture medium. This is a prerequisite for their hydrogel structures if they should be used in medical applications.
In conclusion, the experimental setup as well as the description of the experiments and the presented results are fine and well done. However, the main experiments to prove their claim that Thai SF-based hydrogels are suitable for use as a bioink are missing and thus the manuscript needs major revision.
Reviewer 3 Report
The authors are advised to address the following when they revise the manuscript:
- If there a structural difference between regular silk firoin and Thai silk firoin?
- Tables 1 and 3 could be combined.
- Tables 2 and 4 could be combined.
- Does the addition of STS, DMPG and glycerol interfere with the analysis of FTIR spectra? Should these molecules be removed before analysis?
- Will the mechanics of printing change the gelation time?
- Ratio of SF to glycerol at 3:1, w/w (Table 1)?
- How the degree of recovery was calculated (Table 4)?
Round 2
Reviewer 2 Report
All comments have been edited accurately and satisfactorily. however, especially in the new text passages marked in red by the authors, a grammar and spelling check is necessary again. Thus, I recommend accaptance after minor revision.